# Learning Portable Skills by Identifying Generalizing Features with an Attention-Based Ensemble

## Abstract

The ability to rapidly generalize is crucial for reinforcement learning to be practical in real-world tasks. However, generalization is complicated by the fact that, in many settings, some state features reliably support generalization while others do not. We consider the problem of learning generalizable policies and skills (in the form of options) by identifying feature sets that generalize across instances. We propose an attention-ensemble approach, where a collection of minimally overlapping feature masks is learned, each of which individually maximizes performance on the source instance. Subsequent tasks are instantiated using the ensemble, and transfer performance is used to update the estimated probability that each feature set will generalize in the future. We show that our approach leads to fast policy generalization for eight tasks in the Procgen benchmark. We then show its use in learning portable options in Montezuma's Revenge, where it is able to generalize skills learned in the first screen to the remainder of the game.

## 1 Introduction

In recent years reinforcement learning has outperformed humans in many Atari games (Mnih et al., 2015), learned to play world champion level Go (Silver et al., 2017) and mastered many robot manipulation tasks (Levine et al., 2016; 2018). While these achievements are undeniably impressive, they are in simulated and controlled environments stripped of many of the complexities humans face in everyday life. For reinforcement learning to be viable in real-world applications, the ability to scale to large, high-dimensional environments is crucial. Hierarchical reinforcement learning (Barto & Mahadevan, 2003) is a promising approach to achieve this scalability through the use of high-level skills that abstract away the detail of low-level action. The most popular hierarchical RL framework is the options framework (Sutton et al., 1999)), which models abstract actions as consisting of three components: a set of states from which execution can begin, a policy which specifies how the option executes, and a set of states where execution ceases.

To fully realize the promise of the options framework, learned options should ideally be easily reused, or *ported*, to new tasks and environments (Konidaris & Barto, 2007). The core difficulty here is that, in practice, an option will be first learned in a small number of specific instantiations—possibly just one—without foreknowledge of the circumstances under which it will be applied again in the future. In such cases there may be many state features over which the first instance(s) of the option could be successfully defined, but which will not support reuse. For example, a single option to open a door might be equally well-defined using features describing the door's location in a global map, or features describing the location of its handle relative to the agent, but only the latter will generalize to new doors. This problem is exacerbated by the fact that *all three* components of the option must *simultaneously* function in new instantiations, or the option will fail.

We therefore propose to learn portable options by identifying sets of state features that support generalization. We adopt the transfer learning setting (Taylor & Stone, 2009), where the goal is to learn on a number of source tasks and perform well on target tasks with minimum re-training. We introduce a method where an agent uses an attention-based ensemble (Kim et al., 2018) to learn a collection of diverse feature sets that each individually maximize performance on the source task. Subsequent option instantiations are evaluated for success, and the results are used to update

the agent's estimate of the probability that each feature set will generalize to future tasks. These probabilities in turn govern which feature sets are used in new option instantiations.

We begin by showing how to learn a portable policy using an ensemble, and demonstrate that it leads to fast learning on eight games from the Procgen generalization benchmark (Cobbe et al., 2020). The ensemble is then extended to learn portable classifiers that represent initiation and termination sets. We combine the resulting portable policy, initiation classifier, and termination classifier methods to learn portable options in Montezuma's Revenge, where our method enables an agent to generalize skills learned in the first room to all the others.

## 2    BACKGROUND AND RELATED WORK

We consider the episodic reinforcement learning setting. An agent operates in a Markov Decision Process (MDP) with state space $S$ and action space $A$. The transition probability $p(s_{t+1}|s_t, a_t)$ is the probability of transitioning from state $s_t$ to state $s_{t+1}$ with action $a_t$. At each time step, the agent receives a scalar reward defined by the reward function $r(s_t, a_t)$. The goal of the reinforcement learning agent is to maximize the cumulative discounted reward over an episode by finding a policy $\pi(a|s)$ that selects actions at each step.

The options framework by Sutton et al. (1999) extends the MDP framework by creating temporally extended abstract actions known as options. An option $o$ is defined by a three-tuple $(I_o, \pi_o, \beta_o)$. The initiation set, $I_o : S \to \{0, 1\}$, is the set of states in which option $o$ can initiate. The termination set, $\beta_o : S \to \{0, 1\}$ is the set of states in which option $o$ successfully terminates. The option policy $\pi_o : S \to A$ is a controller that transitions the agent from states in $I_o$ to states in $\beta_o$.

Options in their most basic formulation offer no guarantees of being portable. However, Konidaris & Barto (2007) argue that if the inputs to an option retain the same semantics (introducing the notion of an agent-space) across option instances, these options will be reusable. There are many works focused on building derived input spaces with transferable semantics. Konidaris & Barto (2007) showed that an agent-centric representation, analogous to the egocentric space (the space surrounding the agent (Klatzky, 1998)), would be sufficient for many tasks, especially in the robotics domain. The successor features (SF) and generalized policy improvement (GPI) framework proposed by Barreto et al. (2018) creates a space derived from successor features leading to reusable skills. However, these skills only adapt to changes in the reward function, not the transition dynamics. Gupta et al. (2017) learn skills in an invariant feature space which enables generalization across morphollogically different robots. However, the invariant feature space cannot generalize across tasks. Hausman et al. (2018) propose learning a separate embedding space which can be used to parameterize discovered skills. They show this successfully generalizes these parameterized skills across tasks but make the assumption that the agent has access to a collection of different tasks which are used to learn the embedding space. This embedding space results in portable policies, but does not address the main issue of over-fitting to the few instances an option is defined over. Topin et al. (2015) transfers options between different object-oriented MDPs, but are confined to discrete-domains. Our work instead focuses on building portable options in high-dimensional and continuous domains. We also differ from these approaches by using diversity to build portable options. We propose to learn a diverse set of state features—each of which maximizes rewards—to both define the option policy and identify the portable initiation and termination sets.

One way of maintaining a set of features is by building an ensemble. Ensemble methods train multiple learners on the same task, resulting in a combined model which performs better than each individual model. The quality of the ensemble depends on the quality of each individual and the diversity among them. The attention-based ensemble proposed by Kim et al. (2018) is a deep learning framework, originally intended for deep metric learning, that encourages diversity in the feature embeddings learned by each member in the ensemble. The objective for the ensemble is to minimize a combination of the training objective loss and a divergence loss:

$$L(\{x_i\}) = \sum_m L_{\text{train},(m)}(\{x_i\}) + \lambda_{\text{div}} L_{\text{div}}(\{x_i\}). \tag{1}$$

Here, $\{x_i\}$ is a set of training samples and $L_{\text{train}}$ is the training loss of the learning objective for the $m$-th learner. The divergence loss, $L_{\text{div}}$, is weighted by $\lambda_{\text{div}}$ and defined as

$$L_{\text{div}}(\{x_i\}) = \sum_i \sum_{p,q} \max(0, m_{\text{div}} - d_{\mathcal{Y}}(B_p(x_i), B_q(x_i))^2) \qquad (2)$$

where $d_{\mathcal{Y}}$ is a distance metric in the embedded space, $m_{div}$ is a margin, and $B_p(x_i), B_q(x_i)$ is the embedding output of learners $p$ and $q$. Optimizing the total loss $L(\{x_i\})$ incentivizes learners to individually perform well on the training task while learning maximally different feature embeddings.

## 3 PORTABLE POLICIES WITH AN ATTENTION-BASED ENSEMBLE

We propose to achieve option portability through ensembles. By learning multiple diverse feature sets, we increase our chances of finding a portable one. We then condition the option's policy, initiation classifier, and termination classifier on the portable features, such that the resulting option generalizes to subsequent tasks.

In this section, we focus on learning a portable policy. The agent learns to solve a task in the flat reinforcement learning setting, where only the policy needs to be learned, without the need for initiation and termination sets. We later extend this framework to options in Section 4.

### 3.1 ENSEMBLE OF POLICIES

The main challenge of learning an ensemble of policies is learning an ensemble of feature sets and identifying a portable one. We use an attention-based ensemble architecture to learn $N$ distinct feature embeddings, which in turn are input to $N$ individual reinforcement learners. Figure 1 shows an architecture diagram for an ensemble with $N$ worker policies, each attending to a different set of features.

Each feature-learner consists of three components: a spatial feature extractor, an attention module, and a global feature extractor. The spatial and global feature extractor are shared among all the learners, but each learner has an individual attention module. The attention module encourages each learner to attend to a different aspect of the input, thereby promoting diversity. The input is passed through the shared spatial feature extractor first to obtain the spatial feature embedding $\phi$. This embedding is then passed to each attention module to obtain the attention mask, which is multiplied with the spatial feature embedding to form an attention embedding $\eta_i$. Finally, each attention embedding is passed through the global feature extractor to get the final feature embedding $\mu_i$. Each individual worker policy is conditioned on the features derived from the corresponding member of the ensemble.

The ensemble minimizes the loss defined in Equation 1 with $\lambda_{train}$ defined by the objective loss of the reinforcement learning algorithm. Minimizing this loss encourages the agent to attempt to learn a good policy using each of the feature sets, while keeping feature sets different from one another. In this section, we use Proximal Policy Optimization (PPO) (Schulman et al., 2017) as the reinforcement learner, but our method is compatible with any reinforcement learning algorithm.

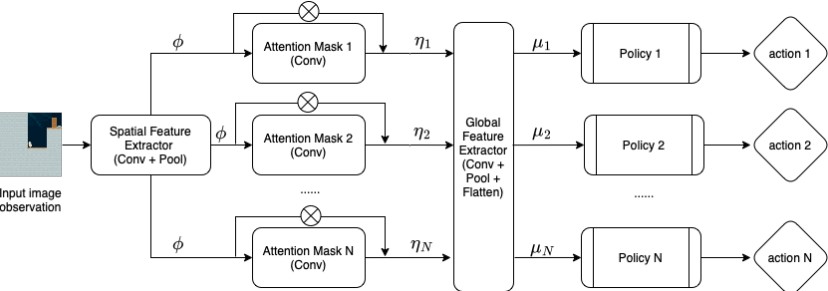

Figure 1: Architecture for learning an ensemble of policies using Attention-Based Ensemble.

Each episode, a single ensemble member, known as the *leader*, is selected to govern action selection and all members are trained on the experiences collected. The choice of leader has a big impact on

the ensemble performance when transferring to new tasks. If the leader is a portable policy, then the ensemble will still be able to perform well during transfer. Otherwise, the ensemble will likely fail. Since we repeatedly select a new leader every episode, it is natural to frame this as an $N$-armed bandit problem (Sutton & Barto, 2018). We can view each choice of leader as a bandit-action and the cumulative reward earned by the leader that episode the bandit-reward. We choose the leader using an Upper Confidence Bound (LAI, 1985) method to balance exploration and exploitation. At timestep $t$, we select leader $L_t$ as follows:

$$L_t = \arg\max_i \left[ R_t^{\text{cum}}(i) + c\sqrt{\frac{\ln t}{N_t(i)}} \right]. \tag{3}$$

$R_t^{\text{cum}}(i)$ is the cumulative reward of ensemble member $i$ at time $t$, $N_t(i)$ denotes the number of times that $i$ has been elected leader prior to time $t$, and constant $c$ controls the degree of exploration. The first part of Eq. 3 can be understood as exploiting good arms, and the second part exploring others.

## 3.2 PROCGEN RESULTS

We showcase the effect of the attention-based ensemble on Procgen (Cobbe et al., 2020), a popular benchmark for evaluating generalization in reinforcement learning. Procgen is a suite of procedurally generated game environments, each offering many levels. Each level is a unique instance of the game, differing in appearance—for example background colour or object shape—but represents the same underlying game mechanism. We evaluated our approach on 8 games: Bigfish, Coinrun, Dodgeball, Heist, Jumper, Leaper, Maze and Ninja[1]. Following Cobbe et al. (2020), we choose PPO (Schulman et al., 2017) as the reinforcement learning algorithm. The training loss $L_{train}$ is the PPO loss detailed in Appendix B.2.

To provide intuition on the effect of the attention modules, we visualize the attention masks at a single time-step of a 3-policy ensemble trained on the game Coinrun in Figure 2. Areas of high attention (yellow) are accentuated by the attention module, while areas with low attention (dark purple) have little effect on the final feature embedding space. Each ensemble member attends to different aspects of the image, each of which represents a different set of features.

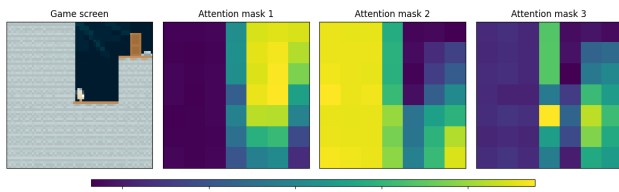

Figure 2: Visualization of the attention masks of a 3-policy ensemble trained on Coinrun, with intensity represented by color. Each attention mask focuses on a different set of features.

We adapt the training scheme of Procgen to better align with the transfer learning paradigm. For each game, we randomly select 20 training levels. The ensemble trains for 500k steps on each level sequentially, maintaining knowledge obtained in previous levels. To evaluate, we compare the average within-level performance across the 20 levels. In Figure 3, we plot the discounted episodic reward achieved on each game, comparing 4 ensemble agents with different ensemble sizes 1, 2, 3, and 5. The baseline agent with only one policy in the ensemble is equivalent to a PPO agent, because the divergence loss in Equation 2 is zero and the attention-based ensemble architecture is equivalent to the feature extraction layers in the PPO. For six out of the eight games, multi-member ensembles outperform the 1-member ensemble, showing better skill transfer across levels. This demonstrates that portable policies can be learned with an ensemble.

To confirm that the improved performance results from learning portable features instead of having an ensemble of policies, which can bring benefits such as better exploration (Osband et al., 2016), we repeat the experiment with the same ensemble of policy approach but with no attention-ensemble

---

[1]In consideration of computing resources, we set the difficulty level to easy

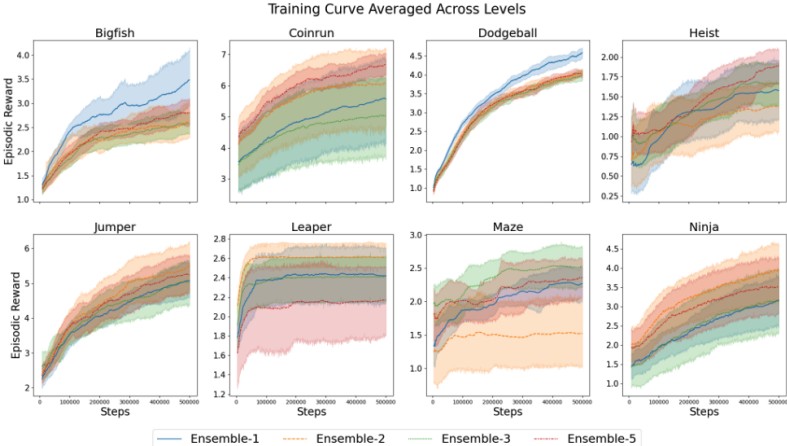

Figure 3: Learning curves of different ensembles of policies on 8 Procgen environments. The 4 different agents have an ensemble of 1, 2, 3, and 5 policies, respectively. On six of the environments, multi-member ensemble agents are able to outperform the baseline agent. (10 random seeds; error bars denote standard deviation.)

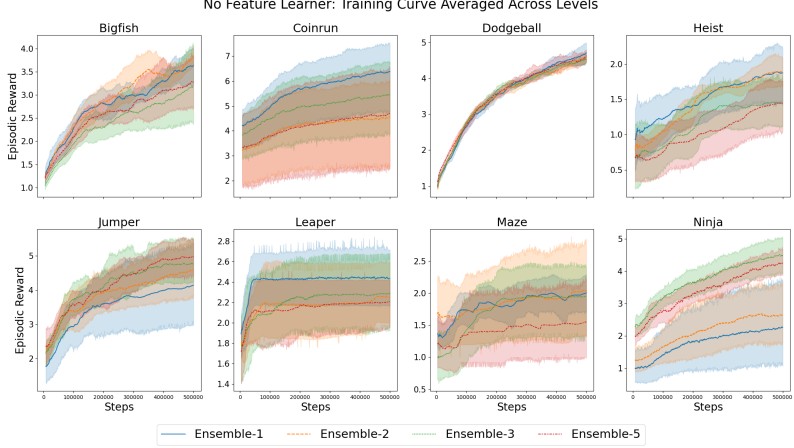

Figure 4: Ablation study of ensemble of policies without attention-ensemble feature learner on 8 Procgen environments. The 4 different agents have an ensemble of 1, 2, 3, and 5 policies, respectively, but all policies share the same learned feature. (6 random seeds; error bars denote standard deviation.)

feature learner. All policies in the ensemble will share a single learned feature. The results in Figure 4 show that performance decreases without learning portable features for five out of eight games.

We note that the optimal ensemble size varies for each game. This is an expected result because of the differing complexities of the environments, which result in different numbers of possible useful feature sets. When the ensemble size is too small, the learned feature representation may not be fine-grained enough to generalize. On the other hand, an excessive number of ensemble members can also degrade performance by spending too much time exploring policies that do not generalize, and not enough time exploiting ones that do. We also draw attention to the importance of the exploration constant $c$ in the UCB algorithm (Equation 3). If $c$ is too small, there is not enough exploration when learning to choose the leader, essentially collapsing the ensemble agent to a 1-member ensemble.

# 4 PORTABLE OPTIONS WITH AN ATTENTION-BASED ENSEMBLE

The results from Section 3.2 show that agent policies can be generalized with an attention-based ensemble. However, a portable option also requires a portable initiation and termination set. We now extend the attention-based ensemble to an ensemble of classifiers that can be combined with the ensemble of policies to learn a portable option. We first go over the construction of the ensemble of classifiers in Section 4.1, and then explain how to combine it with an ensemble of policies to build the portable options in Section 4.2.

## 4.1 ENSEMBLE OF CLASSIFIERS

Similar to the intuition in Section 3, we learn portable classifiers with an ensemble. We use the attention-based ensemble to learn an ensemble of $N$ classifiers, each trained on a unique set of features. This increases the number of feature sets considered when learning the initiation and termination sets, some of which will generalize.

Each classifier in the ensemble learns the set of states that make up the initiation and termination set. The attention-based ensemble learns $N$ diverse embedding spaces which are input to the classifiers. The ensemble is trained using the objective function described by Equation 1, where the training loss $L_{train}$ is the classification loss when predicting whether a state is in the initiation or termination set (here, binary cross entropy), with labels from the instantiating option's classifier. In this paper we use a two layer perceptron as the classifier, and train an ensemble of $N = 8$ classifiers.

Because not all classifiers in the ensemble are portable, we weight each classifier's prediction by the agent's belief of its portability. In Section 3, we modeled this problem as an $N$-armed bandit problem because only one policy can control action-selection during an execution and so successful learning requires exploration. However, when querying the classifiers, we receive predictions from all ensemble members, and so there is no need to explore. As such we can replace the $N$-armed bandit model with a simpler Bayesian update rule

$$P(\theta_i|D_i) = \frac{1}{B(k+\alpha, n-k+\beta)} \theta_i^{k+\alpha-1}(1-\theta_i)^{n-k+\beta-1} \tag{4}$$

where given dataset $D_i$, $k$ out of $n$ classifications are positive. $\theta_i \in [0,1]$ is the prediction from classifier $i$ and $\alpha$ and $\beta$ are selected to represent initial belief over portability. A full derivation of this update rule can be found in Appendix A.

## 4.2 PORTABLE OPTIONS

We build a portable option using an ensemble of classifiers for the initiation and termination sets and an ensemble of policies for the option policy. The training scheme of portable options is shown in Figure 5. The portable option is first trained on the initial task instance (Figure 5 top green box). The agent then explores the environment for possible new instantiations of the portable option (Figure 5 right red box). Finally, the new option instantiation is used to update the portable option after being validated (Figure 5 left blue box). Instance discovery and validation of different instantiations can occur concurrently as the agent continues to interact with the environment.

During *discovery* the agent uses the portable initiation set to identify new states in which the option can be used, or instantiated. Once a new possible instantiation is identified, the portable option policy is executed. If the option execution is successful (as shown in Figure 5), a new instantiation of the option is created. Note that there is no supervision signal for successful executions of an option when it is ported to new tasks because there is no guarantee that the reward function will align with the option termination. As such, it is possible to incorrectly identify new option instantiations if either both the initiation and termination sets fire on false positive states or, due to stochasticity in the environment, the successful option execution cannot be reliably replicated. If the portable option is updated using these incorrectly identified instantiations the option will degrade over time.

Therefore, we use the *validation* stage to refine discovered instantiations before updating the portable option. During this stage the agent learns *Markov classifiers* that characterize the initiation and termination sets of this single option instantiation, and is not portable. These Markov classifiers remove the need for the agent to rely on the possibly unstable portable initiation and termination sets by learning the true initiation and termination sets of one task instance. Unreliable

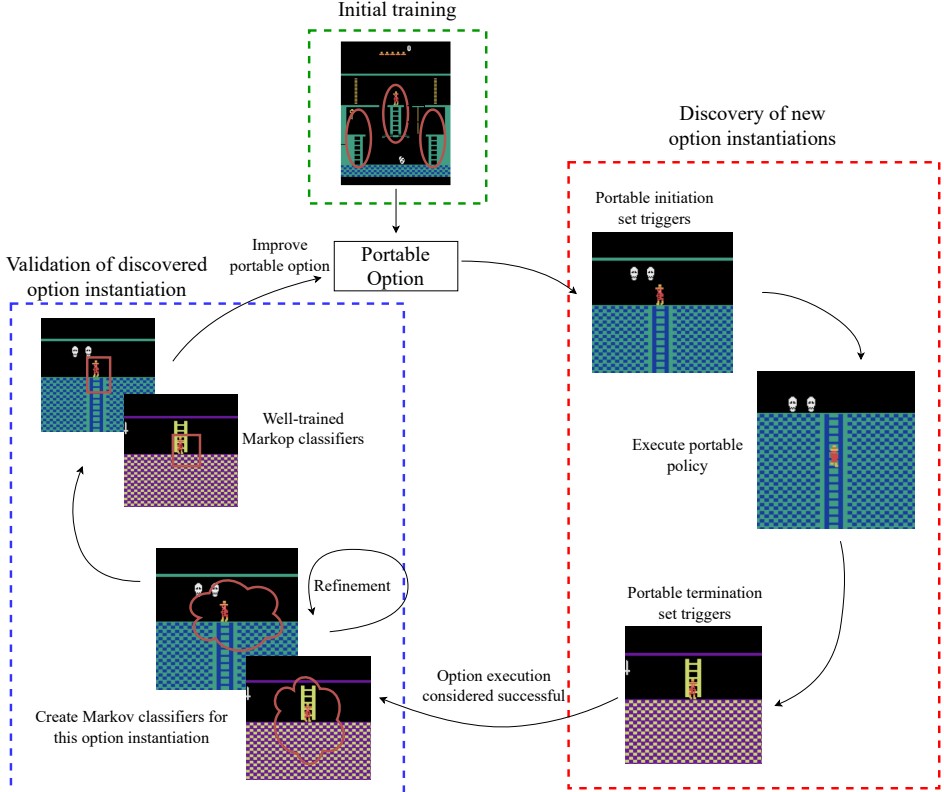

Figure 5: After a portable option is learned the agent continues to explore the environment, finding new possible instantiations of the option. If a valid instantiation is found, this data is used to further improve the portable option. This figures shows this process for a single successful option instantiation.

option instantiations will "disappear" as the Markov classifiers learn that the initiation and termination are null sets and so the instantiation is not used to update the portable option. This ensures that new instantiations are replicable. Finally, the portability belief over classifier ensemble members for the initiation and termination sets is updated.

Before updating the portable option, we impose a check to ensure that the Markov classifiers are "well-trained". We expect a low classification loss for true positive states and, therefore, do not use instantiations with a classification loss above a chosen threshold to update the portable option. This is because, if a feature set is portable, true positive states should be almost indistinguishable from that of previously seen states the classifier deemed positive.

After the *validation* phase, valid option instantiations are used to improve the portable option. This is done by further training the portable initiation and termination sets on samples from the *Markov classifiers*. The portable policy, however, continues to train during the *validation* stage.

### 4.3 MONTEZUMA RESULTS

In this section we validate portable option in the Atari game Montezuma's Revenge, in which the player must navigate a character through multiple rooms filled with enemies, obstacles, and treasures. We define three options that can be utilized in multiple rooms across the game in Table 1. The initiation and termination sets for these options are hand-labeled and only used for initial training (see Figure 5) and evaluation but could in principle be learned by any skill discovery method (e.g. Mannor et al. (2004); Şimşek & Barto (2004); McGovern & Barto (2001)).

The portable option is initially trained on the first room of Montezuma's Revenge, then trained to generalize on subsequent rooms. The portable option receives a stack of four $56 \times 40$ images around

| OPTION | INITIATION | TERMINATION | DESCRIPTION |
|--------|------------|-------------|-------------|
| Climb Down Ladder | The agent is on or above a ladder. | The agent is on a platform. | The agent climbs down the ladder to the next valid platform. |
| Walk Left | The agent can move left without dying. | The agent has reached the left edge of a platform or a ladder. | The agent moves left until the platform ends or a ladder is found. |
| Move Left of Enemy | The agent is to the right of an enemy. | The agent is to the left of an enemy. | The agent must navigate to the left side of the enemy without dying. We define a separate option for each enemy because they require different policies. |

Table 1: A description of options used in Montezuma's Revenge. These options are selected because they should be generalizable to many instances in the game.

the player as input. During initial training, we allow the ensemble of classifiers and ensemble of policies to train until convergence, using the hand-designed initiation and termination sets to provide labels for the classifiers. We then generalize the learned option by training on a collection of other relevant task instances in multiple other rooms across the first level of the game, without hand-designed information. For example, after initially training to descend the ladders of the first room, we place the agent at the top of each ladder in subsequent rooms, and continue training the option on the new ladders. The portable option trains on each new task 100 times through creating Markov classifiers, updating its beliefs, and training the ensemble of policies.

While after initial training the agent is required to determine when an option execution was correct without guidance from the environment, we still need a way to quantify option generalization to validate our approach. We therefore use hand-designed initiation and termination sets for validation and record the final distance from the true termination after option execution, normalized by the full option distance. A perfect execution would result in a distance of 0. If the agent does not attempt the option due to the initiation set not triggering, the distance is 1. If the agent dies during the option execution, ending the episode, the recorded distance is again 1.

We record the option performance after initial training and after each subsequent training task (see Figure 6). This is done by running the portable option on all defined tasks and reporting the performance of the option on previously seen tasks and unseen tasks separately. The performance on previously seen tasks is an indicator of whether the option has deteriorated while performance on unseen tasks shows the option's ability to zero-shot generalize. During evaluation, the portable option attempts each task 50 times and cannot make updates to its portability beliefs or ensemble of policies and classifiers. Additionally, no Markov classifiers are created. We perform this experiment with a multi-head ensemble which has 8 attention heads for the initiation and termination sets and 3 attention heads for the option policy. The experiment is repeated on a single-head ensemble which uses one head for all option components and so does not learn diverse features.

Figures 6a and 6b show that the multi-head ensemble outperforms the single-head ensemble. Additionally, as more tasks are seen, the multi-head ensemble's ability to zero-shot generalize improves, indicating that the diverse features learned by the portable option helps generalization.

For Climb Down Ladder, task 4 and all tasks after task 6 require the agent to learn to avoid an enemy, adding complexity to the initial training task as making contact with an enemy is considered a failed execution. While the ensemble is able to complete these tasks before they are seen during training, it is clear that training on these tasks heavily deteriorates the original option. After all tasks are seen, the option fails on all previous tasks. This is because the ensemble loses confidence in all classifier ensemble members as the agent fails to execute the option. However, while all tasks require the agent to descend a ladder, it can be argued that ladders with enemies may require a different option. Therefore, we likely need a mechanism that allows the ensemble to decide when a new option instantiation should become a new option, which we leave for future work.

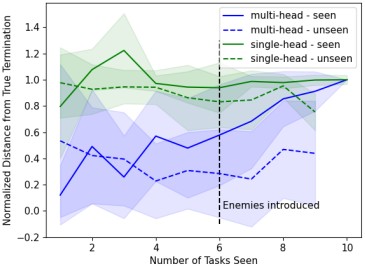

(a) Mean normalized distance from true termination for the Climb Down Ladder option in Montezuma's Revenge averaged over 5 seeds.

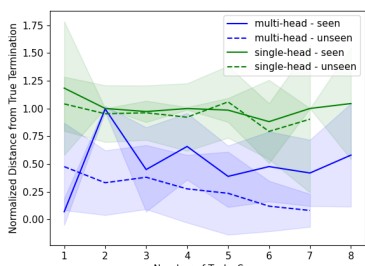

(b) Mean normalized distance from true termination for the Walk Left option in Montezuma's Revenge averaged over 5 seeds.

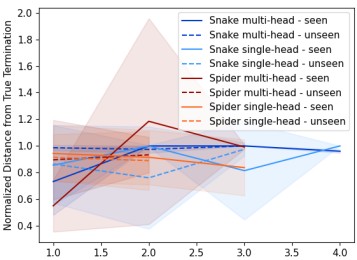

(c) Mean normalized distance from true termination for the Move Left of Enemy option for spiders and snakes in Montezuma's Revenge averaged over 5 seeds.

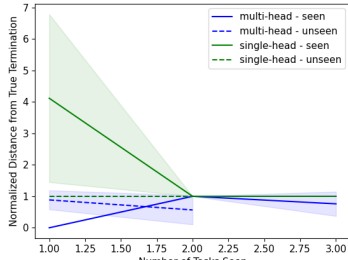

(d) Mean normalized distance from true termination for the Move Left of Enemy option for rolling skulls in Montezuma's Revenge averaged over 5 seeds.

Figure 6: Mean normalized distance from true termination for all Montezuma's revenge experiments.

We see a similar trend in the Walk Left option, where task 2 takes place in a room with disappearing lasers. Due to the four frame stack, this is a partially observable hazard and so appears unpredictable to the agent. We do see, however, that the option is able to recover from this task, with overall performance on all seen tasks decreasing over time.

Avoiding the spider and snake enemies is harder than avoiding the rolling skull enemy. This is evident in Figures 6c and 6d where the multi-head ensembles perform better on the rolling skull compared to the snakes and spiders. However, we see again that the multi-head ensembles deteriorate over time, occasionally performing worse than the single-head ensembles. Similar to Climb Down Ladder, the multi-head ensembles lose confidence in all classifiers in the ensemble if the agent fails the option. This shows that the ability to identify valid option executions is critical to obtain portable options.

## 5 CONCLUSION

In this work, we considered the problem of learning portable options. We proposed the use of an attention-based ensemble that learns multiple, minimally-overlapping feature sets to increase the probability of learning a portable feature set. We used this ensemble to learn a portable policy which we tested on the Procgen benchmark. On six out of eight games we found that the ensemble of policies improved performance, leading to faster generalization across levels. We then extended the attention-based ensemble to classifiers, learning portable initiation and termination sets. We combined the ensemble of policies and ensemble of classifiers to form a portable option which we tested on the Montezuma's Revenge Atari game. We found that portable options were able to reuse a learned option in unseen rooms. For the portable option to improve over subsequent task executions it is critical to identify valid option executions.

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

## A  Ensemble Belief Update Derivation

Because the output of a classifier $i$, $\theta_i$, is a value between $0$ and $1$, we use a Beta distribution for the prior $P(\theta_i)$

$$P(\theta_i) = \frac{1}{B(\alpha, \beta)} \theta_i^{\alpha-1} (1 - \theta_i)^{\beta-1},$$

where $\alpha$ and $\beta$ are selected to reflect our initial beliefs. Given data $D_i$ — if $i$ made a correct prediction — the likelihood a classifier is successful is modelled as a binomial distribution

$$P(D_i|\theta_i) = \binom{n}{k} \theta_i^k (1 - \theta_i)^{n-k}$$

Using Bayes rule, we update our belief in each ensemble member using

$$P(\theta_i|D_i) = \frac{1}{B(k + \alpha, n - k + \beta)} \theta_i^{k+\alpha-1} (1 - \theta_i)^{n-k+\beta-1}. \tag{5}$$

## B  Architecture and Hyperparameters

### B.1  Attention-Based Ensemble Architecture

The attention-based ensemble has three components, a spatial feature extractor, global feature extractor and attention module. The spatial feature extractor consists of a convolutional layer with 32 channels, kernel size of 3 and stride 1 followed by a max pooling layer with a kernel size of 2. Each attention module is a convolutional layer with 32 channels and has a kernel size of 1 and stride 1. We do not include bias in this layer. Furthermore, the attention outputs are normalized to between 0 and 1 with min-max normalization. The global feature extractor consists a convolutional layer, with 64 channels, kernel size of 3, and stride 2, followed by a max pooling layer with a kernel size of 2. The output will also be flattened into a one-dimensional feature embedding.

For experiments on Montezuma's Revenge, the global feature extractor has an appended Gated Recurrent Unit (Cho et al., 2014) layer with a hidden size of 128. We found having this layer to improve generalization performance.

## B.2 PPO

Our PPO policy network consists of two linear layers, where the output feature size is 256 and 15, respectively. The weights of the second layer are initialized with a (semi) orthogonal matrix, and the bias to 0. The output of the policy is fed into the value network to predict the value. The value network is simply a linear layer with output feature size 1.

The PPO agents are optimized according to the loss function proposed by Schulman et al. (2017). At time step $t$, the loss of the network $\theta$ is the combination of three parts

$$L_t^{\text{CLIP+VF+S}}(\theta) = -\mathbb{E}_t[L_t^{\text{CLIP}}(\theta) - c_1 L_t^{\text{VF}}(\theta) + c_2 S[\pi_\theta](s_t)].$$

The clipped objective is

$$L_t^{\text{CLIP}}(\theta) = \mathbb{E}[\min(r_t(\theta)\hat{A}_t, \text{clip}(r_t(\theta, 1 - \epsilon, 1 + \epsilon))\hat{A}_t)],$$

where $\hat{A}_t$ is an advantage estimator. The value function loss is

$$L_t^{\text{VF}} = (V_\theta(s_t) - V_t^{\text{targ}})^2.$$

Finally, $S$ denotes an entropy bonus.

## B.3 Q Network

For experiments on Montezuma's Revenge, the ensemble of policies has the same structure of an attention-based ensemble followed by policy networks. Each policy network is a q function, parameterized by a neural network with two linear layers. The first layer outputs embeddings of dimension $64$, and the second layer outputs q values of dimension $18$. The policy selects actions using an epsilon-greedy scheme, where the epsilon parameter is being decayed linearly through time.

The updates to the q function uses samples collected and stored in a replay buffer. We weight each sample with Hindsight Experience Replay (Andrychowicz et al., 2017).

## C  Environment Settings

### C.1  Procgen Environment

For all the experiments in Procgen, we use 8 parallel environments to speed up learning. The discount factor is set to $\gamma = 0.999$. We always center the image observations on the game character.

We normalize the reward function using the Parallel Algorithm Chan et al. (1982).

The results in Figure 4 report the discounted episodic return for every time step. Furthermore, the episodic rewards are smoothed over a sliding window of 100 episodes.

### C.2  Montezuma's Revenge Environment

We use a frame stack of four previous time steps, where each frame is a crop of the game screen with the rectangular area around the player sprite, as input to the portable option. The size of the agent space is $56 \times 40$ pixels centered around the player. During option training in the first room, we ignore the environment reward, and instead only reward $+1$ when an option succeeds, and $0$ otherwise.

While the portable option is trained using these cropped images, the Markov classifiers are trained on the player's position in the game.

## D  Experiment Hyperparameters

Table 2: PPO ensemble of policies hyperparameters.

| HYPERPARAMETER | VALUE |
| --- | --- |
| $c_1$ | 0.5 |
| $c_2$ | 0.01 |
| Clip (likelihood ratio & value function) | 0.2 |
| Maximum $L_2$ norm for gradient clipping | 0.5 |
| Agent update period | 256 |
| Batch size | 1024 |
| Number epochs per gradient update | 3 |
| Learning rate | $5 \times 10^{-4}$ |
| $\lambda$-return | 0.95 |
| Replay buffer size | $10^5$ |
| Bandit exploration weight $c$ | 500 |
| Embedding distance function | Euclidean distance |
| $m_{div}$ | 1000 |

Table 3: Q-function ensemble of policies hyperparameters.

| HYPERPARAMETER | VALUE |
| --- | --- |
| Hidden layer output size | 128 |
| Discount rate | 0.9 |
| Target update interval | 10 |
| Exploration algorithm | Linear decay epsilon greedy |
| Initial epsilon | 1.0 |
| Final epsilon | 0.01 |
| Epsilon decay steps | $10^6$ |
| Ensemble size | 3 |
| Replay buffer size | $10^5$ |
| Batch size | 32 |
| Learning rate | $2.5 \times 10^{-4}$ |
| Bandit exploration weight $c$ | 100 |
| Embedding distance function | Euclidean distance |
| $m_{div}$ | 1 |

Table 4: Ensemble of classifiers hyperparameters.

| HYPERPARAMETER | VALUE |
| --- | --- |
| Ensemble learning rate | $10^{-4}$ |
| Classifier learning rate | $10^{-2}$ |
| $\alpha$ | 100 |
| $\beta$ | 10 |
| Maximum loss | $3\times$ training loss |
| Initial embedding training epoch | 100 |
| Initial classifier training epoch | 50 |
| Batch size | 32 |
| Markov classifier interaction minimum | 100 |
| Maximum time steps before option timeout | 50 |
| Embedding distance function | Euclidean distance |
| Markov Classifier Architecture | Two-class support vector machine |
| $m_{div}$ | 1 |

