# OpenReview forum: "Learning Portable Skills by Identifying Generalizing Features with an Attention-Based Ensemble"
_ICLR.cc/2023/Conference — Submitted to ICLR 2023_

### Official Review · Reviewer_Q9hT · 2022-10-23

**Confidence:** 4
**Correctness:** 2
**Technical Novelty And Significance:** 3
**Empirical Novelty And Significance:** 2
**Recommendation:** 3

**Clarity, Quality, Novelty And Reproducibility:**

The writing is clear in introduction and the first part of the approach section. The writing in the later part of the approach section (that explains extension of the idea to the option setting) as well as in the experimental evaluation however is hard to follow and lacks visualizations.

The quality of the experimental evaluation is severely lacking.

The approach is novel to the best of my knowledge and an interesting idea.

**Strength And Weaknesses:**

# Strengths

- the problem of learning portable options is very important: the whole point of learning options is to “port” them to different tasks, but often options trained on observations from one task overfit to spurious features and then don’t generalize well to another — approaches like the one proposed here, that aim to address this problem, are of high relevance to the community

- the intuition of the proposed (learning ensembles with disjoint feature sets) is novel to my knowledge and intuitive

- I like the structure of the paper, in which the authors first explain their idea in the context of policy learning and only afterwards extend it to the more complex case of option learning (which additionally requires to learn initiation and termination sets) —> disentangling the complexity in this way makes it much easier to follow the description of the method


# Weaknesses

The main weakness of the submission is the experimental evaluation of the proposed approach. The paper proposes an approach for option learning, but never evaluates whether the learned options are actually useful for downstream learning. There are no comparisons to prior works and no ablation studies of the components of the proposed approach. There is no qualitative analysis of the learned feature representations, even though they are the main novelty proposed in the approach.

The experimental analysis will need substantial improvements to verify the usefulness of the approach.

(1) Evaluations on using the options for downstream task learning should be added.

(2) A comparison to (a) prior option learning works and (b) the exact same setup as the proposed approach but without the ensemble learning part should be included.

(3) Detailed analysis experiments of the many moving parts of the proposed approach should be added: how do different number of ensemble elements or different regularizing coefficients influence the learned options?

(4) Qualitative results and visualizations should be added, to show what features the ensemble identifies as transferrable, in what situations transfer works well (compared to baselines), in what situations the proposed approach fails etc

The current experimental evaluation does not allow to properly judge the proposed approach and, due to a lack of qualitative results, it is hard to understand why some of the trained options fail or why their performance *degrades* over training.

-----
Apart from the experimental evaluation, there are a few additional weaknesses:

(A) The description of the extension of the ensemble approach to learning options (instead of policies) was hard for me to follow, particularly I had trouble understanding the second-to-last paragraph in section 4.2 — some intuitive examples for options and feature sets, e.g. from Montezuma’s revenge, and a figure visualizing the approach could help here.

(B) The proposed objective encourages the learned representations between ensemble elements to be disjoint — it seems to me that this can make optimization tricky since there are many local optima in which important features are “trapped” within disjoint feature sets of different ensemble members, making me wonder how hard it is to tune this approach on tasks that require the interplay of many features for policy learning.

(C) More generally, like many option papers, the proposed approach has many moving parts (learning ensembles of policies, initiation and termination classifiers, additional regularization terms with coefficients that require tuning, bandit-optimization of which policy from the ensemble to pick for execution). This suggests that it will be hard / require a lot of tuning to apply this approach to complex problems. Again, this is generally true for many option learning approaches, but the paper introduces additional complexities that go beyond conventional option learning and exacerbate the problem.

(D) The related work section misses some references to relevant work on learning state abstractions for generalization in RL. See e.g. Amy Zhang’s McGill PhD thesis from 2021 for an overview of relevant works.


# Questions


- Can we learn a single embedding space shared between initiation and termination classifiers and policy? This could substantially reduce the number of learned components and required tuning.

- Fig 3 shows that the number of ensemble elements needs to be tuned separately for each task. Is there a good way to choose the number of ensemble elements without just trying all of them?


**Summary Of The Paper:**

The paper proposes an approach for learning portable options by training and ensemble of options and enforcing that the ensemble members learn feature representations that attend to different parts of the observation. The approach then learns to determine the feature set that generalizes best across environments to choose the most portable option. The paper demonstrates learning of portable policies in ProcGen and portable options in Montezuma’s revenge.

**Summary Of The Review:**

I am a bit torn about this submission. I really like the tackled problem and the approach seems interesting to me. There are some doubts about the empirical performance of the approach which I have outlined in “weaknesses” above. These would require a solid experimental evaluation to address. However, the experimental evaluation of the current submission is severely lacking. Thus I do not recommend acceptance of the paper in it’s current form and suggest the authors extend the experimental evaluation and resubmit their paper at ICLR or another fitting venue.

---

> ### Author Response · Authors · 2022-11-10
> **Regarding downstream learning**
>
> Thank you for your review, comments and suggestions. We are adding more experiments and rewriting section 4 to improve clarity.
>
> With respect to the effect of these options on downstream learning, unfortunately for Montezuma's Revenge the hyperparameter tuning requires a large amount of time (1 week for a single run) and compute resources. Additionally the game still suffers from sparse rewards even if we had perfect options, requiring exploration techniques and tuning. While we might be able to have these experiments complete for a camera-ready version we would not be able to complete this before the end of the rebuttal period.
>
> Once again thank you for your time and review!

---

> > ### Comment · Reviewer_Q9hT · 2022-11-24
> > **Thanks for your response!**
> >
> > Thanks for your reply! Judging from your responses it seems that the added experiments will make the paper stronger. Regarding the comparison to prior work: I don’t have a full overview of prior option learning approaches, so it is possible that this is the first paper to do option learning from images, but even then it would be good to include comparison to an adaptation of prior option learning work to image observations (eg switch out the encoders of the policy/termination networks if that is possible).
> >
> > Since most of the crucial experiments were only added during the rebuttal and some (like downstream task learning) are still lacking, I still do not recommend acceptance for this submission, but I encourage the authors to resubmit once the experiments and updates to the paper are complete.

---

> ### Author Response · Authors · 2022-11-17
> **Revised Submission**
>
> Thanks again for your suggestions.
>
> > “A comparison to (a) prior option learning works and (b) the exact same setup as the proposed approach but without the ensemble learning part should be included”
>
> To the best of our knowledge, our work is the first to learn portable options in image-based domains so there is no meaningful comparison to prior work. However, as you suggested, it is important to compare our proposed approach to the setting without the ensemble part. We have added additional experiments for that which can be found in the updated version of the paper.
>
> We have repeated the Montezuma experiments using a single head for all portable option components which can be found in the paper or here:(([ladder](https://ibb.co/sHgzSGx), [snake/spiders](https://ibb.co/DkCvvsz), [rolling skull](https://ibb.co/3YJV9Rm), [walk left](https://ibb.co/6F9zvhK))). We found that the single-head attention fails to generalize. We have also changed our evaluation metric to the normalized distance from the true termination, normalized by the true option distance (ground truth initiation to ground truth termination).
>
> > “Detailed analysis experiments of the many moving parts of the proposed approach should be added”
>
> We have added additional ablation studies for the attention-based ensemble in the paper. Particularly, we want to highlight the new experiment that shows the improvement in performance on the Procgen benchmark indeed comes from having learned portable features, instead of coming from increased exploration from having a policy-ensemble. In the experiment, we kept the ensemble of policies, but removed the ensemble feature learner so that all policies share a single learned feature set. The results can be found in the paper or [here](https://ibb.co/fnbrWrc). The plot shows that only having a policy ensemble, without learning a portable feature set, results in lower performance gains, and is in fact sometimes detrimental.
>
> > “Qualitative results and visualizations should be added, to show what features the ensemble identifies as transferrable, in what situations transfer works well (compared to baselines), in what situations the proposed approach fails”
>
> We have performed additional experiments to analyze the situations in which the attention-ensemble approach works well and fails. In particular, in the Procgen benchmark, our experimental results show that 2 out of 8 games the ensemble fails. We attribute this failure to the lack of meaningful portable features in the game: in these games, the screen changes frequently over time to display different game objects that cannot be meaningfully ported. Features that are portable (e.g. the pixels around the controlled agent in the game) is not in a fixed location on the screen and so the attention mask is not able to pick them out.
>
> We have also separated the ensemble performance on seen and unseen tasks which gives additional insight to what causes the portable option to deteriorate. This deterioration is due to the heavy penalties that are applied when the agent dies during an option execution, with drops in performance correlating with tasks that include enemies which end the episode prematurely if the agent collides with them.
>
> > “ The description of the extension of the ensemble approach to learning options (instead of policies) was hard for me to follow”
>
> We have submitted an update to the paper in which we rewrote section 4 and added additional visualizations which clarifies our proposed method further.
>
> > “The proposed objective encourages the learned representations between ensemble elements to be disjoint — it seems to me that this can make optimization tricky since there are many local optima in which important features are “trapped” within disjoint feature sets of different ensemble members, making me wonder how hard it is to tune this approach on tasks that require the interplay of many features for policy learning.”
>
> We would like to point out that while the ensemble is encouraged to learn diverse feature sets, this is not a hard requirement. Each ensemble member is trying to learn the feature set that will maximize individual performance. If there are beneficial features they may appear in multiple attentions as long as the overall feature set is not the same. Additionally, the amount of shared features allowed can be tuned as is mentioned in the original attention-based ensemble paper (Kim et al., 2018).
>
> > “The related work section misses some references to relevant work on learning state abstractions for generalization in RL.”
>
> Thanks for making the connections to relevant work, we have added more references in the related works section. Our work differs from the state abstraction literature because we do not learn a fixed state abstraction of the whole state space, instead learning a portable set of features relevant to a single option.
>
> Thank you once again for your time and please let us know if this did not address your concerns!

---

### Official Review · Reviewer_8Rro · 2022-10-24

**Confidence:** 3
**Correctness:** 2
**Technical Novelty And Significance:** 3
**Empirical Novelty And Significance:** 1
**Recommendation:** 3

**Clarity, Quality, Novelty And Reproducibility:**

Clarity: the motivation is clear and enjoyable to read. The method section and the experiment section are really difficult to understand.
Quality: the motivation and the premise are clear. But I had a hard time evaluating the validity of the method and the evaluation.
Originality: learning reusable feature set for options framework is important and novel.

**Strength And Weaknesses:**

Overall I like the premise of the paper. Options should be able to port to other environments or different parts of the state space, as opposed to only function in the state distribution that it’s trained on. I also applaud that the paper chooses to evaluate its method on a pixel-input domain as opposed to more “convenient” environments where each input dimension is already meaningful.

However, I have to admit that I have a hard time appreciating the method itself and the conclusion drawn by the evaluation. I also had a hard time understanding the method itself, despite multiple attempts at reading the section. Below I will describe my comments in detail.

Writing quality
The structure of the paper is clear, and the introduction is enjoyable to read. However, the method section is difficult to understand, especially section 4.2 and 4.3 and their connections wit the previous subsections. Here is a list of points that I do not fully understand
I understand why it’s difficult to evaluate the initiation condition, but if the method assumes a reward function is given, why can’t it use the reward function to determine the termination set?
The third paragraph of 4.2 says “If the initiation and termination set are both incorrect during the same option execution, the option will appear to succeed but will have an incorrect instantiation”. I understand false positives are likely, but the sentence appears to state that the option is guaranteed to succeed if both classifiers are wrong. Why is that?
I’m generally confused about the role of the markov classifier. The paragraph says the markov classifier “learns the initiation or termination set of a new option instantions”, What does that mean?
How does markov classifier allow the option to “return to a specific instantiation”? Does it reset the simulator status?
In the 5th paragraph, the “classification loss” is used to determine policy success. The classification loss of which classifier? And why is loss used, not the prediction? And why is a true positive state “almost indistinguishable from the previously seen state the classifier deems positive” since the invariant feature should only be a subset of the entire space?
How is the portability estimate in 4.3 used by the classifier ensembles concretely?
In section 4.4, what exactly is the “true determination state” in eq 5? What is the “true” meant to contrast?

My second major comment is regarding the conclusion drawn in the experiments in general, especially in Section 3.2. The paper seems to imply that the improved performance is solely due to the reusable feature set extracted via the ensemble feature learning scheme. However, another factor that may play a critical role is the ensemble policies themselves. It is widely known that ensemble could aid exploration. Thus to correctly attribute the performance gain, It would be critical to ablate this factor and evaluate the ensemble RL policy without the feature learner. Similar counter arguments can be made for all later experiments, as they all rely on the same ensemble RL strategy.

Finally, the paper has many moving parts and little ablation studies. The ambiguous evaluation metrics that entirely relies on the learned classifier makes the experiment results less clear. I believe the paper will benefit tremendously from a set of experiments using domains that support option-level skill evaluation with ground truth initiation and termination set. For now, it is really difficult to understand the validity of the conclusion drawn from the experiments.


**Summary Of The Paper:**

The paper tackles the problem of learning reusable option policies to achieve cross-task generalization using reinforcement learning. It describes a method for learning feature spaces that can be reused by an option-like policy for (1) determine the next action to take (2) whether it is executable and (3) whether it should terminate. The core technical component is an ensemble feature learner (Kim et al., 2018) that aims to discover a set of feature extractors that are both diverse and optimizes some objective. In this paper, the ensemble feature learner is used to discover features for an ensemble of RL-based option policies for optimizing task rewards. The same ensemble feature learner is also used to learn an ensemble of initiation set and termination set classifier. The core component of learning reusable policies using feature ensemble learner is evaluated on a set of Procgen environments. The full method (including the initiation and termination set classifier) is evaluated on the montezuma’s revenge domain, where different rooms are considered different environments.


**Summary Of The Review:**

Overall I really like the premise and the motivation of the paper. However, the writing / clarity is lacking and the experiment evaluation is insufficient.

---

> ### Author Response · Authors · 2022-11-10
> **Some clarifications**
>
> Thank you for your thorough and thoughtful review. We are in the process of adding additional experiments, improving the evaluation metric and rewriting section 4 for clarity, but we wanted to provide some initial clarification below first.
>
> > “I understand why it’s difficult to evaluate the initiation condition, but if the method assumes a reward function is given, why can’t it use the reward function to determine the termination set?”
>
> Our goal is for options to be as reusable as possible. If a reward function was used to determine the termination set, the termination set would not be guaranteed to be portable to new tasks that has different reward functions that might not correctly reward option completion. In Montezuma's Revenge (a screenshot of the first screen can be found in the paper or here), the option that climbs down ladders appears to be a useful, self-contained option that can be reused at multiple points throughout the game. However, the reward in the game is received when the agents collect the key, for example, and cannot ge used to train the climb-down-ladder option. Additionally, we aim for our method to be compatible with any option discovery method to learn the options initially. Relying on reward function may limit the number of compatible option discovery techniques.
>
> > "I understand false positives are likely, but the sentence appears to state that the option is guaranteed to succeed if both classifiers are wrong. Why is that? I’m generally confused about the role of the markov classifier."
>
> To use option instantiations to update the portable option, we need to determine when an option execution is successful. However, we have no additional information from the environment that can indicate whether the option was correctly used. So, we deem any option execution that begins in the initiation set (i.e. the initiation classifier triggers) and ends in the termination set (i.e. the termination classifier triggers) when following the option policy successful. However, this definition could have two possible issues
> The option happened to succeed because of stochasticity in the environment and such execution success is not replicable by the agent.
> The initiation and termination sets both triggered on false positives states. This is the situation you are mentioning. In this setting, the option execution will appear to be successful, but actually accomplishes a different task than originally intended. The Markov classifier is used in this situation to keep track of how often a single option instantiation succeeds. This will be clarified further in the updated section 4.
>
> Finally, thanks for bringing up the need to ablate the factor of potential exploration benefits of the policy ensemble. We are currently running ablation studies on the Procgen benchmark. However, for the experiments on Montezuma’s Revenge, we don’t think the ensemble method will introduce potential benefits in exploration. This is because the initiation set classifier must be triggered before option execution, and the initiation set is a classifier that does not benefit from nor provide additional exploration. Similarly, the termination set is also a classifier that does not relate to exploration.
>
> Thank you again for your feedback!

---

> ### Author Response · Authors · 2022-11-16
> **new updates**
>
> We have updated the paper to include a rewrite of section 4 and additions of visualizations for clarity. We believe this will resolve the confusion surrounding our proposed method, including the detailed points you kindly listed.
>
> Secondly, we have finished an ablation study to separate the potential exploration benefits of the policy ensemble. The results can be found in Section 3.2, which we will also summarize here. We ran new experiments where the ensemble of policies all share a single learned feature, thus removing the learned portable feature. The results can be found in the paper and also at this link ([https://ibb.co/fnbrWrc](https://ibb.co/fnbrWrc)). The plot shows that only having a policy ensemble, without learning a portable feature set, does not give as much performance improvement, and is in fact sometimes detrimental. Therefore, we can conclude that the performance improvement indeed results from learning portable feature sets.
>
> Thanks again for your time, and please let us know if our response did not fully address your concerns.

---

### Official Review · Reviewer_rc41 · 2022-10-27

**Confidence:** 4
**Correctness:** 3
**Technical Novelty And Significance:** 2
**Empirical Novelty And Significance:** 2
**Recommendation:** 3

**Clarity, Quality, Novelty And Reproducibility:**

The paper is quite easy to follow up until section3. However, I found section4 which brings together the entire approach with the ensemble of classifiers and markov classifier, as well as the various specific conditions used in the montezuma experiments to be more convoluted.
Due to the complex nature (multiple moving parts) of the overall system, it might be difficult to reproduce.


**Strength And Weaknesses:**

Strengths
1. Generalization through learning diverse representations
Using an ensemble of policies each of which focus on different parts of the input, and then selecting among them using a bandit-style objective to find the most robust policy is an intriguing idea. There has been work that focuses on learning the right representations of the environment to enable better generalization [1], and this paper approaches the same problem by instead encouraging diverse representations for multiple learners (enforced using distance in the embedded space), and then selecting for the most robust one.

Weaknesses -

1. Complexity of approach, specificity of Montezuma experiments, relation to exploration algorithms

The overall approach has a large number of moving parts - the policy ensemble, the classifier ensemble, the markov classifier. For the experiments included in particular in the paper (Montezuma's revenge), the initiation and termination conditions need to be specified by hand for the classifiers. Even the options are hand-specified, and not learned for the task (the authors seem to indicate that there is further training for these options, but this isn't clear). Also evaluation is done on specific tasks within the game (rolling skull, spider etc). Contrast this to results in self-supervised exploration. Go-Explore [2] was able to enable an agent to learn how to play Montezuma's revenge without any of these constraints (hand specified option/initiation/termination conditions) and without any rewards, and keep generalizing across different levels also without any restriction to the particular task considered.


2. Sensitivity of ensemble size for policy ensemble -
 Even for the policy ensemble results on the ProcGen environment, the policy ensemble size needs to be tuned for each environment, and performance does not generally improve with more policies in the ensemble. The authors argue this is because 'too much time is spent exploring policies that do not generalize'. However, there is no analysis that shows this to be the case. It might be difficult for practitioners to adopt this approach if there is heavy reliance on tuning this hyper-parameter. The results on the ProcGen environments show that no single choice for ensemble size performs consistently (they each are close to the 1-member ensemble case in atleast one environment).


[1]: Eysenbach, Ben, Russ R. Salakhutdinov, and Sergey Levine. "Robust predictable control."
[2] Ecoffet, Adrien, et al. "Go-explore: a new approach for hard-exploration problems."

**Summary Of The Paper:**

The paper proposes an approach for learning policies that can enable better generalization across tasks. This is done using an ensemble of learners, each of which uses attention to focus on a different part of the feature space, and then using bandit-exploration to select a particular policy. Further, the authors include an ensemble of classifiers to account for initiation and terminations sets, and argue that the combination of the classifiers and the policies give rise to transferable options. Generalization results are included on ProcGen, and Montezuma's revenge (across multiple levels).

**Summary Of The Review:**

Overall, I am not in favor of acceptance since the full presented system is evaluated for particular choices of options/initial/final conditions and specific tasks, while we have have algorithms from the self-supervised learning literature like Go-explore which can keep exploring and generalizing (for the same environment considered in this paper). The idea of learning diverse features to find one that's robust to new tasks seems interesting but performance is very dependent on ensemble size as described previously, and this will hinder widespread adoption.

---

> ### Author Response · Authors · 2022-11-16
> **Clarifications**
>
> Thank you for your review.
>
> As you pointed out, our work proposed many methods to deal with hard problems in generalization: the policy ensemble learns a portable policy using Upper Confidence Bound; the classifier ensemble learns portable initiation and termination sets; the Markov classifiers learn to refine successful option instantiations and remove false positive ones. While our method consists of many parts, as is common in deep hierarchical reinforcement learning [1] [2] [3], we have shown through experiments on Montezuma’s Revenge that they work well together and can effectively generalize options from the first screen to the rest of the game.
>
> With regards to our experiments on Montezuma’s Revenge, we hand-specify the tasks but the options are learned through continual learning. We would like to clarify that the initiation and termination sets are only hand-specified in the first screen where the option is initially defined and not yet generalized, and is only used for evaluation later on and not for training. This hand-designation process exists only to ensure that the option is initially well-defined and can meaningfully be transferred to solve similar tasks in the game. Were the options discovered and not specified, the agent could learn a skill that is not applicable anywhere else in the game, rendering the evaluation of generalization meaningless. However, given an environment where meaningful generalization is possible, our method can be paired with existing option discovery methods to learn options, removing the need for hand-defined options. While we think such extensions are meaningful, it is outside the scope of this work.
>
> While Go-Explore [4] can achieve an incredible performance on Montezuma’s Revenge, it addresses the problem of exploration. Our work is in the orthogonal direction of skill generalization. In fact, our method can be used together with any exploration algorithm, such as Go-Explore. Furthermore, we would like to note that our goal in this work is not to outperform state-of-the-art algorithms to solve Montezuma’s Revenge. Rather, we aim to investigate how options can be transferred and reused on new tasks, and use Montezuma’s Revenge as an evaluation environment. To the best of our knowledge, our method is the first that is able to create portable options in high-dimensional and continuous domains.
>
> Admittedly, the need to tune the ensemble size of the policy-ensemble is a necessity of our proposed method. However, this is a hyperparameter of the network architecture and hyperparameter tuning is widely needed and expected in deep learning, even though undesirable. We don’t think the reliance on hyperparameter tunning will hinder the adoption of our method. Furthermore, even though the optimal ensemble size needs to be tuned, we have shown on 8 Procgen games that having an attention-ensemble is helpful for transfer regardless of ensemble size. To address your concern about the lack of analysis on over-exploration, we are currently running new experiments and will add the results to the paper.
>
> Thanks for making the connection to [5]. However, that work does not concern hierarchical reinforcement learning, and we believe our directions to be orthogonal.
>
> Finally, we have rewritten section 4 and added more visualizations for clarity. We believe it resolves the concern for interpretability.
>
> Thank you for your time, and please let us know if our response did not fully address your concerns.
>
> [1] Lyu, D., Yang, F., Liu, B. and Gustafson, S., 2019, July. SDRL: interpretable and data-efficient deep reinforcement learning leveraging symbolic planning.
> [2] Zhao, D., Zhang, L., Zhang, B., Zheng, L., Bao, Y. and Yan, W., 2020, July. Mahrl: Multi-goals abstraction based deep hierarchical reinforcement learning for recommendations.
> [3] Peng, X.B., Berseth, G., Yin, K. and Van De Panne, M., 2017. Deeploco: Dynamic locomotion skills using hierarchical deep reinforcement learning.
> [4] Ecoffet, A., Huizinga, J., Lehman, J., Stanley, K.O. and Clune, J., 2021. First return, then explore.
> [5] Eysenbach, B., Salakhutdinov, R.R. and Levine, S., 2021. Robust predictable control.

---

### Decision · Program_Chairs · 2023-01-20

**Decision:**

Reject

**Justification For Why Not Higher Score:**

The paper was missing important empirical validation in the initial submission. Based on the initial reviews, it has become stronger, but would need to be re-reviewed in detail. In addition, the authors have not shown the usefulness of the learned options for downstream tasks.

**Justification For Why Not Lower Score:**

N/A

**Metareview: Summary, Strengths And Weaknesses:**

The paper focuses on the problem of learning generalizable skills in multi-task reinforcement learning settings. The authors propose an ensemble approach, which is empirically validated on the procgen benchmark and the Atari game Montezuma's revenge.

Reviewers appreciated the importance of the problem tackled here - a key argument behind options is generalization, yet this is a hard and underexplored area. The paper proposes a promising idea, and reviewers find the approach intuitively appealing. They also commend the fact that evaluation was performed on challenging domains with visual observations.

At the same time, initial reviews noted several concerns, some of which have been addressed by the authors during the rebuttal phase. The algorithm was found to be quite complex, and the clarity of describing the approach needed improvement. Reviewers raised questions about the hand-specified components (initiation, termination as specified on screen 1 of MR), and need for hyperparameter tuning (ensemble size). Additional concerns were raised regarding a lack of ablations to confirm the role of individual components of the approach, comparison to prior work, and a demonstration of the usefulness of the learned generalizable options to downstream tasks.

The authors responded to several of these concerns, clarifying the approach in section 4, and providing additional experiments that covered the requested ablations. The paper is assessed to be in much stronger shape than before. At the same time, concerns remain. First, the additional experiments are a substantial and necessary addition to the paper, and should be re-evaluated in a thorough set of reviews. Second, empirical validations such as the usefulness for downstream tasks have not been added. In sum, the paper has improved during the rebuttal phase, but is not yet ready for publication at the current stage.